# Large Deflections of Thin-Walled Plates under Transverse Loading—Investigation of the Generated In-Plane Stresses

**DOI:** 10.3390/ma15041577

**Published:** 2022-02-20

**Authors:** Gilad Hakim, Haim Abramovich

**Affiliations:** Faculty of Aerospace Engineering, Technion—Israel Institute of Technology (IIT), Haifa 32000, Israel; ghakim@outlook.com

**Keywords:** thin-walled plate, large deflection, von Kármán equations, in-plane (membrane) stress, finite element analysis, boundary conditions, movable edges, immovable edges

## Abstract

Thin-walled plates subjected to transverse loading undergoing large deflection have been the topic of a large number of studies. However, there is still a lack of information about the nature and the distribution membrane stresses generated under large deflections. The purpose of this paper is to calculate and display the distribution of the generated stresses and the respective deflections on the entire rectangular plate area. Finite element analysis results for thin-walled plates with aspect ratios of 1, 2 and 5, on movable and immovable edges simply supported and clamped boundary conditions are clearly visualized. The distribution of the normal and shear stresses enables a good understanding of the plate critical points locations. It was found that strong tensile and compressive membrane stresses exist at various points near the plate edges, creating potential failure hazards.

## 1. Introduction

The behavior of flat plates subjected to various in-plane and out-of-plane loading has attracted much attention due to its technological importance. This vast problem can be divided into many sub-problems depending on the parameters to be investigated, such as plate thickness, perimeter shape, material properties, small vs. large deflections, and shear deformability. A thin-walled plate made of isotropic material, loaded by transversal pressure is one of the classical problems in elasticity, with an enormous number of studies being written on the subject (see, e.g., [1,2,3]).

Plates undergoing small deflections that do not the exceed fraction of the plate thickness (usually less than 0.2·*t*, *t* = thickness), have shown linear behavior with good satisfactory analytical solutions as described in [1,2,3]. However, for larger deflections of a few times the thickness and higher, a non-linear behavior exists caused by stretching of the mid-surface of the plate, thus increasing the transverse plate stiffness. The load–deflection graph line ceases to be linear as a function of the transverse pressure, and the whole stress distribution is a function of the out-of-plane and the in-plane deformations. In 1910 Theodore von Kármán [4] made a major breakthrough for the plate large deflections problem. He published a set of non-linear differential equations that describes the large deflection behavior, considering the in-plane deformations and stresses. They are known in the literature as Föppl–von Kármán equations, named after August Föppl [5] and Theodore von Kármán [4], or in short, von Kármán equations, which have the following form:(1)$$\frac{\partial^{4}F}{\partial x^{4}}+2\frac{\partial^{4}F}{\partial x^{2}\partial y^{2}}+\frac{\partial^{4}F}{\partial y^{4}}=E\left[{\left(\frac{\partial^{2}w}{\partial x\partial y}\right)}^{2}-\frac{\partial^{4}w}{\partial x^{2}\partial y^{2}}\right]$$
(2)$$\frac{\partial^{4}w}{\partial x^{4}}+2\frac{\partial^{4}w}{\partial x^{2}\partial y^{2}}+\frac{\partial^{4}w}{\partial y^{4}}=\frac{q}{D}+\frac{t}{D}\left[\frac{\partial^{2}F}{\partial y^{2}}\frac{\partial^{2}w}{\partial x^{2}}-2\frac{\partial^{2}F}{\partial x\partial y}\frac{\partial^{2}w}{\partial x\partial y}+\frac{\partial^{2}F}{\partial x^{2}}\frac{\partial^{2}w}{\partial y^{2}}\right]$$
where *E* is the plate’s Young’s modulus, *w* is the out-of-plane deflection, *t* is the thickness of the plate, *q* is the transverse uniform pressure and *D* is the flexural bending presented as
(3)$$D=\frac{Et^{3}}{12\left(1-\nu^{2}\right)}$$
with ν being the Poisson’s ratio of the plates material and *F* is the Airy function [3] defined as
(4)$$\sigma_{xx}=\frac{\partial^{2}F}{\partial y^{2}}\;,\;\sigma_{yy}=\frac{\partial^{2}F}{\partial x^{2}}\;,\;\tau_{xy}=-\frac{\partial^{2}F}{\partial x\partial y}$$

It is interesting to note, as pointed out by Bakker et al. [6], that Equations (1) and (2) are a simplification of Marguerre [7] equations for plates having initial imperfections and subjected to in-plane and transverse loads (the initial imperfection is taken as zero in Equations (1) and (2)).

## 2. State of the Art

Unfortunately, von Kármán’s equations set turned out to be a very difficult and cumbersome to solve. To date, there are still no closed form analytic solutions for rectangular flat plate that satisfy both the differential equations and the boundary conditions. Nevertheless, many approximate and numerical solutions were published (see typical examples in [6,8,9,10,11,12,13,14,15,16,17,18,19,20,21,22,23,24]). The approximated methods presented in the literature would suggest solutions, usually having severe limitations. Most of them are not easy to use and the non-linear nature of the plate hardening effect in large deflection mode is not evident.

The following literature review tries to establish the state of the art for rectangular and circular plates undergoing large deflections due to lateral loading.

Browsing the literature reveals that the non-linear behavior of flat plates using von Kármán’s two equations was mainly investigated for the transverse deflections of the plate, with the in-plane (the membrane) generated stresses being less discussed. NACA had allocated a lot of efforts to investigate the issue by publishing technical reports in the years 1941–1951 (see typical reports in [8,9,10,11,12,13,14,15]). Various methods were employed such as multiplying double Fourier series results in quadruple series with a large number of terms leading to numerical tables and graphs with the membrane stresses in the *x* and *y* directions being calculated at the mid-point of a square plate and mid-point edges and the shear stresses set to zero [9]. The numerical work was further enhanced in an experimental study on aluminum-squared plates [10]. Another interesting study is presented in [11] for a clamped plate having an aspect ratio of 1.5 and undergoing large deflections. Their results differ only by 3% from an infinitely long plate, thus implying that long plates should be treated as infinitely long and the in-plane stress distribution along lines parallel to the edges going through the plate center not changing significantly.

Following the studies presented by Samuel Levy [9,10,11,12], Wang [13,14] solved the large deflection problem by employing two finite differences schemes, the successive approximations and relaxation method to yield a good comparison with Levy’s results. He considered an all-around immovable clamped plate and all-around simply supported movable boundary conditions. Both square and rectangular plates are presented with the in-plane stresses being calculated at three points, center of the plate, long edge mid-point and short edge mid-point, without indicating the presence of compression type stress. One should note also the study presented in [15] for sandwich type plates, for which numerical and experimental results are presented and well compared.

In 1954, Berger [16] assumed that the strain energy due to the second invariant of the middle surface strain can be neglected leading to a solution of the von Kármán’s equations set. This neglection would mean that for the large deflections case, the plate’s bending resistance is low, and the plate would behave like a pure membrane. The study presents results for circular and rectangular flat plates for both clamped and simply supported boundary conditions, with deflections and stresses being presented graphically and numerically at certain points on the plate.

In 1969–1970, Scholes and Bernstein [17] and Scholes [18] presented approximate large deflections solutions using energy methods for all around simply supported rectangular plates [17] and all-around clamped plates [18]. A good comparison with experimental results is reported in [17]. They employed Timoshenko’s [3] mentioned idea to divide the loading path into a first part which, would cause bending, and a secondary part leading to membrane stretching, and the load–deflection curve calculated by a finite differences scheme. The clamped case being dealt in [18] present stresses and deflections calculations and comparison with measured results. Maximal values are presented for a pressure-loaded plate to enable efficient design.

Li-Zhou and Shu [19] use the perturbation variational method to solve the large deflections problem of rectangular plates under transverse pressure leading to analytical expression for displacements and stresses. They report a good comparison with available experiments.

Bert et al. [20] also address von Kármán’s equations for orthotropic rectangular plates. The solution is obtained and presented using the differential quadrature method. The boundary condition used in their study was all around simply supported and all around clamped, both immovable. They report deflections, membrane, and bending stresses are in good agreement to known solutions. Values of the stresses are calculated at the plate’s mid-point as a function of the applied transverse load. Yeh and Liu [21] also addressed the issue of approximated analytic solution for the orthotropic von Kármán’s equations. The presented solution leads to an expression for the self-mode frequency. Numerical and graphical solutions are presented only for deflections while the stress distribution being not dealt in the study.

More recent studies, such as by Wang and El-Sheikh [22], present results for the von Kármán’s equations by multiplying Fourier series, getting quadruple sums, and equating similar terms in the results series. The output is a nonlinear algebraic equations system with 1, 3, 4, 6, or 9 equations and unknowns, according to the number of terms taken for the series. This system is solved for every desired point on the plate. For that, the authors use numerical tools based on Generalized Reduced Gradient (GRG) method. They also present a closed form solution for the mid-point deflection using the first term only having the form of *q* = α·*w* + γ·*w*^3^ (*q* = transverse load, *w* = lateral deflection and α, γ = fitted constants). However, there are indications from other sources that the use of only one term is not accurate enough and has serious deviations from the reality. An interesting solution for the Föppl–von Kármán equations set is presented by Bakker et al. [6] using an approximated analytic solution. Thanks to the simplicity of the trial function, the bending and the membrane loading influences are separated, easing the solution process. The results are compared to ANSYS FEA (Engineering Simulation Software, Canonsburg, PA, USA) results with less than 10% difference. Six combinations of boundary conditions and four loading cases make the results presentation rather complex. Stresses are presented using various formulas without any graphical outcome.

Ugural [23] in his book, Ch.10, presents approximate solutions for circular thin plate S-I (simply supported, immovable edge). However, for the solution for a thin rectangular plate he assumes membrane-only stresses (no bending resistance) at the mid-point, with SSSS–I boundary conditions.

Razdolsky [24] also presents approximate solutions for rectangular SSSS–I rectangular plates with deflections and stresses calculated for several aspect ratios. He converts the stress expressions of Levy [9,10,11,12] through minimum potential energy method to computer executable algorithms. His square plate deflection curve is found to be between Timoshenko [3] and Levy [9,10,11,12] curves, while for the stresses no direct comparison is presented.

Turvey and Osman [25] performed numerical analysis with finite differences dynamic relaxation (DR) of square isotropic Mindlin (shear deformable) plates. His results are said to be in “generally good agreement” with Alamy and Chapman (1969) and Rushton (1968), but no comparison is shown.

Paik et al. [26] have developed complex expressions for thin plates large deflection using the Galerkin method. Their example includes both transverse and axial edge compression load. Since the results show only in-plane loads on the edges, it is difficult to compare it to a case without these loads.

Nishawala [27] handles both nonlinear beams and plates. For plates, both movable and immovable edges are displayed. Several other sources are compared for deflections, but without stresses. He suggests a 3rd degree polynomial load–deflection expression for the plate mid-point.

Jianqiao [28] uses both Boundary Elements (BE) and Finite Element (FE) to calculate deflection and mid-point stress for both simply supported and clamped immovable edges. A comparison with Boshton (1970) was made, yielding a good agreement.

Abayakoon [29] studied beams as the main subject, while presenting also plates using a 3rd degree polynomial mid-point deflection expression. A deflection comparison is made to Timoshenko [3] and others. Stresses are simulated for ribs stiffened plates only, which cannot be compared with thin plates.

Seide [30] presents an expression for deflection, but for stresses it is limited to infinitely long plate, which cannot be used for a square plate.

Parker [31] solved the plate problem with finite differences. He presents membrane stress results with good agreement with Levy [9], but less good with Wang [14].

Belardi et al. [32] analyze a circular plate made of shear deformable orthotropic composite materials. While the material has cartesian *XY* orthotropy, the other variables in the analysis are polar. The shear deformations are calculated using the FSDT (first order shear deformation theory). Deflections and rotations are presented. Stresses are considered, but without presented results.

Plaut [33] in a recent study uses Reissner theory for plates, that allows large strains (not to be confused with shear deformation) for circular and annular thin plates with both movable and immovable BCs. Results for various loading cases are presented, but for deflection only. No in-plane stresses are considered.

Finally, Shufrin et al. [34] solved the problem of laminated rectangular plates under large deflections with a semi-analytic method considering the coupling coefficients tension–bending and bending–twisting. The nonlinear partial differential equations are converted to an iterative process of ordinary nonlinear differential equation according to Kantorovich method. The result is large mathematical expressions, calculated and compared to ANSYS FEA (Engineering Simulation Software, Canonsburg, PA, USA) with a good agreement. Several cases of local loads (patch type load) are also demonstrated. In-plane stresses including shear stress are partially given along certain lines, loading arrangements.

Based on the above references, it seems that the membrane stress distribution along a plate under large deflections has been neglected in the literature. Despite the extensive research conducted in this field, there is still lack of graphic pictures that describe the in-plane tensile and shear stresses generated in the plate as a function of the transverse loading. One should remember that under a transverse distributed load, the plate would deform. The general shape of the deformed plate is rather intuitive and easy to predict. However, unlike the deflection shape, the membrane stress fields created within the loaded plate are beyond our natural perception and are generally unknown. This is a real problem when trying to find areas with high tensile or compressive stress.

It is the aim of the present study to investigate the membrane stresses generated by the large deflections regime and present their distribution and critical values accompanied by graphical figures. The behavior of thin rectangular plates having various aspect ratios under transverse constant pressure were calculated for increasing loading parameters. Four types of boundary conditions were applied: all around simply supported, with the in-plane movement being allowed (movable) or restricted (immovable), and all-around clamped edges with movable or immovable edges. The calculations were performed using Siemens Simcenter Femap with Nastran Ver. 2021.1 code (finite element, Siemens, Germany) [35].

The novelty of the present manuscript is the comprehensive display of the in-plane stresses for the entire plate in contrast to other existing studies in which the stresses were presented only for specific points, thus ignoring the complete picture. Moreover, it was found that the high compression stress does not necessarily occur at the edge mid-point, which has not been noticed before. Finally, a somehow surprising find in the form of a complex shape having sharp variations for the tensile and shear stresses near the plate corners was for the first time displayed, which has not been noticed in the literature

One should note that the compressive stresses might have the potential to create local buckling which might be considered as a failure, while the strong tensile stresses can create cracks and cause failure of the loaded plate.

## 3. Materials and Methods

The first model consisted of a thin square flat plate made of isotropic material with linear elastic response. The plate’s dimensions were 6.28 × 6.28 m^2^ (length × width). The plate had a thickness *t* of 12 mm and was made of isotropic plastics Polycarbonate. The material Polycarbonate was a transparent tough elastic polymer that is used in many engineering applications such as aircraft cockpit canopy, safety goggles, compact disks, and greenhouse glazing. The type used in the present study was isotropic, and has a Young’s Modulus of 2.4 GPa, Poisson’s ratio 0.38, density 1200 kg/m^3^, tensile strength 63 MPa, high impact strength, and price of about 2.7 €/kg. This material showed a perfect linear stress–strain behavior in all cases described within the present study.

The *xyz* axes origin was at the plate’s mid-point with *z* being normal to the surface.

The plate was loaded in the *z* direction with an evenly distributed transverse load of *q* = 800 Pa or *q* = 75 Pa (see Figure 1).

Two out-of-plane general boundary conditions were applied: the first one being all around simply supported (transverse deflection and moment being zero and designated as SSSS) and the second one all around clamped (transverse deflection and rotation being zero and short named as CCCC). In addition, in-plane boundary conditions were used: allowing free movement of the plate’s edges in the *x* and *y* directions nicknamed *movable* (M) or preventing this movement in both *x* and *y* directions leading to the case of *immovable* (I), (see Figure 2).

As a result, four combinations of boundary conditions: SSSS–M, SSSS–I, CCCC–M, and CCCC–I were used throughout the present study.

To calculate the non-linear response of the transverse loaded plate, the Siemens Simcenter Femap with Nastran Ver. 2021.1 code [35] was used, enabling a high-resolution visualization of both the deflections and the generated stresses on the plate. The FEA process has three main steps: model preparation, running the solver and post processing. Model preparation steps included various topics: definition of the geometry of the plate including its thickness, definition of its material, meshing the surface using 2D CQUAD4 element (capable of modeling in-plane, bending, and transverse shear behavior), application of the distributed load on the plate surface, and defining the boundary conditions. Note that the CQUAD4 element was a quadrilateral one, with bending and membrane stiffness, and had 4 grid points and 5 Gauss points. The four types of BC were alternatively set on the surface perimeter as:SSSS–M: TZ, designated 3;CCCC–M: TZ,RX,RY, designated 3,4,5;SSSS–I: TX,TY,TZ, designated T;CCCC–I: TX,TY,TZ,RX,RY,RZ, designated F.

Additionally, the FEA required the elimination of all free body DOF (degrees of freedom). Therefore, virtual BC were added, where necessary: plate midpoint may have had TX, TY, designated 1,2 and one edge may have had additional TX or TY. Finally, the type of the analysis was defined, for our case SOL106, which included nonlinear large deflections. The second step was the analyzing of the model constructed in the previous step. The third step, the post processing, contained a very rich set of tools allowing to observe and report any requested feature of the plate performance. For the present application, four output vectors were chosen to display the deformation and contour (color) styles of the results: total translation (deflection), plate X membrane force, plate Y membrane force, and plate XY membrane force. The resulted pictures (2D and 3D) were then saved to be used in the report.

The mesh density had an influence on both the accuracy and the calculation time. In the following convergence study, several models with various mesh densities were tested for deflection and force vs. calculation duration. The highest density 500 × 500 was taken as the reference—100%. The results are shown in the following Table 1 and graph Figure 3:

Using the above results, the 100 × 100 mesh density was selected for the entire work, having a good balance between accuracy and calculation duration, and no significant changes occurred in higher densities.

A nonlinear static analysis with 20 steps was applied, where the transverse load was gradually increased, while deflections and stresses were recalculated at each step. This analysis was repeated for each of the four boundary conditions, described above.

In Femap, the analysis program 10.Nonlinear Static uses SOL 106 [37] which is able to handle many nonlinear situations such as:Large deflection with small strains;Nonlinear stress–strain material response;Material plastic yield;Geometric nonlinearities;Creep behavior;Snap-in mechanism;Physical contact between objects;Thin-shell buckling.

In our case, however, the only nonlinear parameter was large deflection that caused system stiffening due to in-plane stresses that accumulated additional elastic energy, resulting in a nonlinear load–deflection response. The way to handle this nonlinearity was to increase the load in 20 steps, where in each step the program used a linear formulation but performed several iterations, until convergence of the energy and the load was obtained. The program may have added additional steps when the convergence was slow. A complete description of SOL 106 and nonlinear static analysis is available (see the link in the reference list [37]).

## 4. Validation

To validate The FEA used here, a comparison to results presented in [3] S. Timoshenko and S. Woinowsky-Krieger, Theory of Plates and Shells (1959, 1987) 2nd edition, p. 427 was made. A Femap model with the same plate configuration as in [3] was used.

The properties of the thin plate model are:
Size:Length × Width a × a [m] = 1 × 1Thicknessh [m] = 0.006Young’s ModulusE [GPa] = 2.4 (Isotropic material)Poisson’s Ratioν = 0.316Distributed load (uniform)q [Pa] = 871BCsSSSS–I (4 edged Simply Supported, In-plane Immovable)Mesh100 × 100 Plate, Cquad4 elements (Plane stress)AnalysisNonlinear Static, 15 StepsModel run timet [sec] = 35Points A,B,C—are located 5 elements away from the plate edge

The following graph, Figure 4, of nondimensional stress vs. nondimensional pressure q, was adapted from [3] and includes also the present FEA results:

As can be seen from the graph, there is an excellent agreement of the present FEA results with Timoshenko’s results for points A, C, and D, while for point B a maximal deviation up of 16.5% was detected. In view of the excellent agreement for points A, C, and D, and the deviation for point B, it is suggested that an error might occurred on the drawing of the original graph (Timoshenko’s book [3]) and it might have a higher inclination, eventually coinciding with the present FEA results.

The following pictures, Figure 5a,b, present the 2D and 3D views of X direction membrane forces, as calculated by the present FEA code.

One can conclude that the results of the present FEA code are viable and correctly represent the stresses on the plate.

## 5. Results

### 5.1. Square Plate (AR = 1)

Figure 6a–d presents the deformed shape of the plate at a transverse load of 800 Pa, for the four boundary conditions used in the present study. As can be seen also from Table 2, the in-plane boundary conditions (movable vs. immovable) play a major role in stiffening the lateral deflection of the plate. Restricting the in-plane boundary conditions reduces the transverse deflection by a factor of 3.3 for the all-around simply supported (SSSS) boundary conditions or by a factor of 2.7 for the all-around clamped case (CCCC). Note also the large, normalized transverse deflections (see Table 2) experienced by the plate for movable in-plane boundary conditions (26.83 for SSSS and 21.58 for CCCC) in comparison with immovable in-plane boundary conditions (8.13-8.0 for SSSS and CCCC).

Note that the vertical *z* axis was scaled to have a better view of the deflection.

The calculated membrane forces are next presented. Note that as the cross section of the plate is constant, the distribution of the membrane forces also depicts exactly the stresses on the plate (a division of the membrane force by a factor 12,000 would yield the stress in MPa at the same point).

Figure 7a–d presents 2D and 3D views of the *x* and *y* membrane forces’ distribution generated on the plate on SSSS–M boundary conditions due to large transverse deflections. As the investigated plate is a square, it is obvious that the *y* membrane forces map (or the *y* stresses map) (see Figure 7c) has an identical appearance to the *x* membrane forces map (or the *x* stresses map) after 90° rotation in the *xy* plane (see Figure 7a). Therefore, when the *y* membrane stress is sought, one would be referred to the *x* stress map rotated by 90°.

The associated membrane stresses maps for a CCCC–M plate are given in Figure 8, displaying interesting fluctuations along the edges of the plate.

Based on Figure 7a–d and Figure 8a,b, some preliminary observations can be put forward:The maps for the distributions of $\sigma_{xx}$ and $\sigma_{yy}$ membrane stresses are symmetrical relative to both *x* and *y*-axes.The middle plate area encounters tension stresses in both *x* and *y* directions, with moderate changes of the amplitude.At the plate edges, namely at y=±3.14 m high $\sigma_{xx}$ compression stress is visible, with a similar behavior at x=±3.14 m- high $\sigma_{yy}$ compression stress. The presence of compressive stresses on the plate’s edges might lead to local buckling at those areas.At the plate’s corners, very sharp changes in the stress amplitudes are encountered possibly due to the relatively coarse mesh at those locations, as the mesh distribution was kept constant across the plate.The maximal midpoint tensile stress (in both *x* and *y* directions) for the SSSS–M case is 1.684 MPa, while the maximal compression stress on the plate’s edges reaches the value of −10.767 MPa. For the CCCC–M case, those stresses reach the values of 1.518 MPa and −6.674 MPa, accordingly.

The behavior of the plate, while applying immovable boundary conditions is presented by Figure 9a–d.

Unlike the movable boundary conditions, the immovable SSSS–I and CCCC–I present only tensile forces. The $\sigma_{xx}$ and $\sigma_{yy}$ membrane stresses maintain their symmetry relative to both *x* and *y* axes, tension stresses in both *x* and *y* directions exist on the plate middle area, with moderate changes, while in the vicinity of the plate edges at x≈0,y=±3.14 m, high tension values of $\sigma_{yy}$ are present as well as at x=±3.14 m,y≈0, where high $\sigma_{xx}$ is generated by the large deflections of the plate. Note that, approximately at the plate corners, one can find low stress values. The maximal midpoint tensile stress (in both *x* and *y* directions) for the SSSS–I case is 2.095 MPa, while the maximal tension stress on the plate’s edges reaches the value of 2.404 MPa. For the CCCC–I case, those stresses reach the values of 1.993 MPa and 2.037 MPa, accordingly.

The distribution of the shear forces and accordingly the $\tau_{xy}$ stresses are presented in Figure 10a–d for SSSS–M and CCCC–M, Figure 11 (a detail of Figure 10b) and in Figure 12a–d for and SSSS–I and CCCC–I boundary conditions. Note that for a better visualization of the shear stresses at the plate corner, the mesh was increased to 500 × 500 and the result is presented in Figure 11.

Based on Figure 10, Figure 11 and Figure 12, one can observe the following:The $\tau_{xy}$ stress function is anti-symmetric relative to both *x* and *y* axes, namely $\tau_{xy}\left(x,y\right)=\tau_{xy}\left(-x,-y\right)=-\tau_{xy}\left(x,-y\right)=-\tau_{xy}\left(-x,y\right)$. In addition, the function is symmetric relative to both square main diagonals, namely $\tau_{xy}\left(x,y\right)=\tau_{xy}\left(y,x\right),\;\tau_{xy}\left(x,y\right)=\tau_{xy}\left(-x,-y\right)$ and its value along the *x* and *y* axes is zero.Although the boundary conditions for the movable cases (M) require $\tau_{xy}=0$ on the plate edges, the calculated shear stresses near the edges is not zero. This discrepancy might be explained by remembering that the finite element membrane forces are calculated at the element mid-point, which is half width of the element distant from the edge.Very sharp changes of the shear stresses values are encountered at the plate corners for the movable (M) cases, which is similar to the movable (M) cases tensile stress variations depicted in Figure 7b,d and Figure 8b.The maximal values of the shear stresses are located on the two main diagonals of the plate. For the SSSS–M case we get ${\left(\tau_{xy}\right)}_{max}=\pm 3.224\;\mathrm{MPa}$, while for CCCC–M boundary conditions we get ${\left(\tau_{xy}\right)}_{max}=\pm 1.938\;\mathrm{MPa}$. Changing the boundary conditions from movable (M) to immovable (I) drastically reduces the shear stresses, to yield for the SSSS–I a value of ${\left(\tau_{xy}\right)}_{max}=\pm 0.278\;\mathrm{MPa}$, while for the CCCC–I case a value of ${\left(\tau_{xy}\right)}_{max}=\pm 0.174\;\mathrm{MPa}$ was calculated.

### 5.2. Rectangular Plates (AR = 2 and AR = 5)

The influence of the plate’s aspect ratio was next investigated. Two aspect ratios were chosen, AR = 2 and AR = 5, where the *x* direction is the longer edge. The mesh for the AR = 2 was 100 × 200 while for AR = 5 it was 100 × 500. The transverse pressure was reduced to 75 Pa, to keep the deflections within the software limits.

Note that the vertical *z* axis was scaled to have a better view of the deflection.

For reference to the square plate, the distributions of the lateral deflection for plates with AR = 2 and AR = 5 are presented in Figure 13 and Figure 14.

The general shapes of the deflection functions of all cases presented here are rather similar, with some minor changes between aspect ratios and boundary conditions. The shape of the movable (M) case tends to be more oval, while the shape of the immovable (I) case tends to be more rectangular.

The distribution of the membrane forces, leading to the distribution of the $\sigma_{xx},\sigma_{yy},\tau_{xy}$ stresses is presented in Figure 15a–f for a plate with an aspect ratio of AR = 2 on SSSS–M boundary conditions and for a plate with AR = 5 and SSSS–M in Figure 16a–f.

One should note that to obtain the relevant stresses, the calculated membrane forces should be divided by a factor of 12,000 to yield the values in MPa at the same points.

Modifying the in-plane boundary conditions from movable (M) to immovable (I) leads to a different distribution of the membrane forces (or to the distribution of the $\sigma_{xx},\sigma_{yy},\tau_{xy}$ stresses) which is presented in Figure 17a–f for a plate with an aspect ratio of AR = 2 on SSSS–I boundary conditions and for a plate with AR = 5 and SSSS–I in Figure 18a–f.

The distribution of the membrane forces, leading to the distribution of the $\sigma_{xx},\sigma_{yy},\tau_{xy}$ stresses is presented in Figure 19a–f for a plate with an aspect ratio of AR = 2 on CCCC–M boundary conditions and for a plate with AR = 5 and CCCC–M in Figure 20a–f.

Finally, the map shape of the membrane forces, leading to the distribution of the $\sigma_{xx},\sigma_{yy},\tau_{xy}$ stresses is presented in Figure 21a–f for a plate with an aspect ratio of AR = 2 on CCCC–I boundary conditions and for a plate with AR = 5 and CCCC–I in Figure 22a–f.

## 6. Discussion and Conclusions

A flat plate was modelled using the Quad4 element within the Siemens Simcenter Femap with Nastran Ver. 2021.1 finite element code. The plate was loaded by transverse pressure, yielding large out-of-plane deflections. Four types of boundary conditions were investigated, SSSS–M, SSSS–I, CCCC–M, and CCCC–I. The influence of the plate’s dimensions was also investigated by calculating specimens with aspect ratios of 1, 2, and 5.

A clear visualization of the membrane (in-plane) stresses is presented enabling the designer to understand the exact distribution of the generated stresses both on the plate’s edges and on its main loaded area. Table 3 presents a summary of the extremal values of the generated stresses $\sigma_{xx}$, $\sigma_{yy}$, and $\tau_{xy}$.

Based on the investigation presented above, the following conclusions can be drawn:The general shape of the plate deflection is rather similar for all the investigated cases. The various four applied boundary conditions do not significantly change the appearance of the deformed plate.Enabling the in-plane movement of the plate would generate higher membrane stresses for both SSSS and CCCC boundary conditions in comparison with restricting this movement yielding an immovable boundary condition.The stresses generated on the plate due to its large transverse deflections for the clamped cases CCCC–M and CCCC–I, are consistently lower than those on simply supported cases SSSS–M and SSSS–I (see Table 3).Compression stresses would appear on the plate edges for both movable SSSS and CCCC boundary conditions. This should be considered during the design of the plate, to prevent local buckling of the plate.To prevent local buckling of the plate, it is recommended to assure CCCC or SSSS immovable boundary conditions, leading to only tensile stresses.The existence of tensile stresses in the movable cases at relatively high aspect ratio AR = 5, suggests checking how these stresses asymptotically approach zero for the case of infinitely long plate, where the membrane tensile stress must be zero.

## Figures and Tables

**Figure 1 materials-15-01577-f001:**
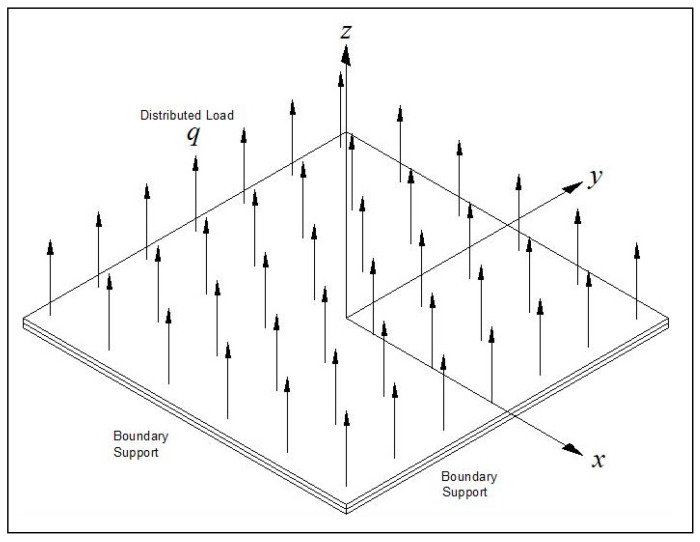
The *xyz* axes and applied pressure.

**Figure 2 materials-15-01577-f002:**
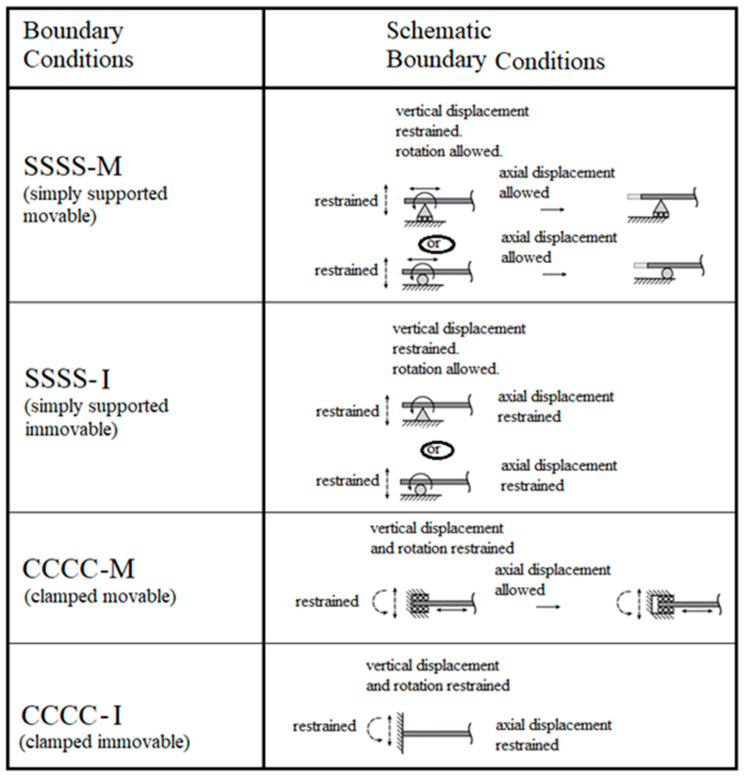
The schematic boundary conditions used in the present study (Adapted from [36]).

**Figure 3 materials-15-01577-f003:**
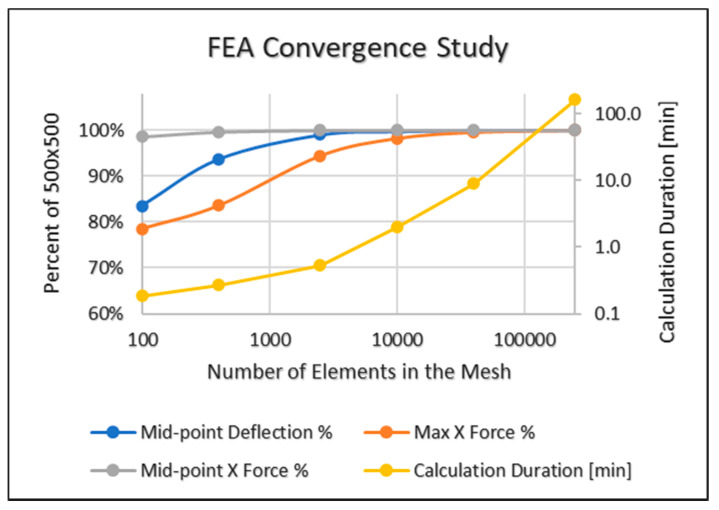
Finite element analysis convergence vs. calculation duration and mesh density.

**Figure 4 materials-15-01577-f004:**
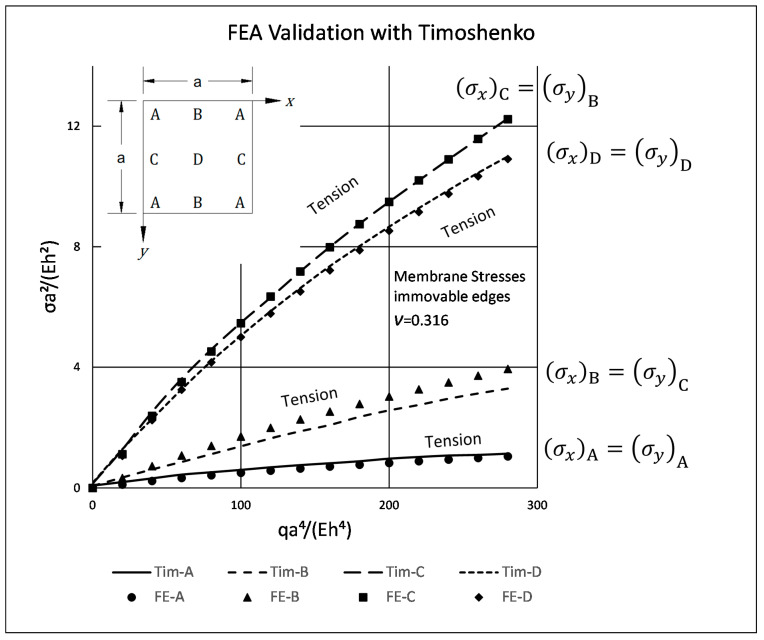
Nondimesional stress vs. nondimensional applied pressure-validation case.

**Figure 5 materials-15-01577-f005:**
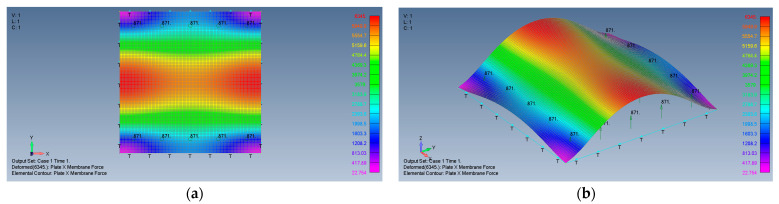
The FEA model 2D (**a**) and 3D (**b**) views of X direction membrane forces.

**Figure 6 materials-15-01577-f006:**
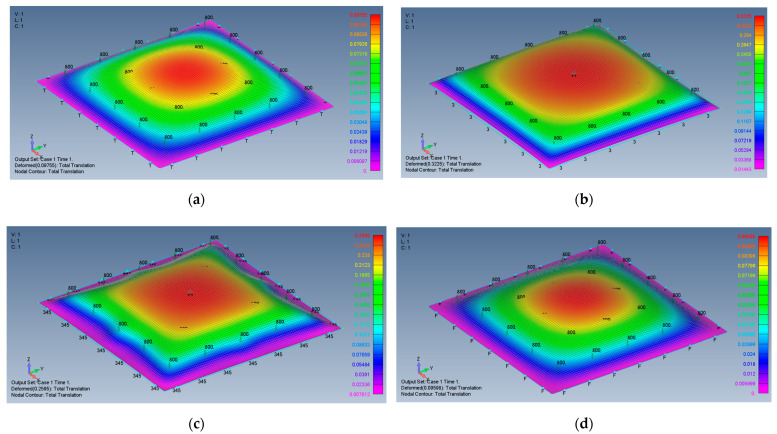
The shape of the transverse deflection for a square plate: (**a**) SSSS–M, (**b**) SSSS–I, (**c**) CCCC–M, (**d**) CCCC–I.

**Figure 7 materials-15-01577-f007:**
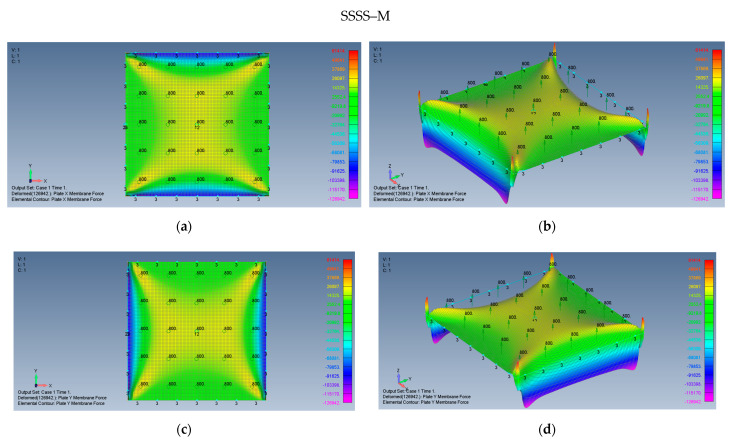
The shapes of the membrane forces for a square plate on SSSS–M boundary conditions: (**a**) top view of the *x* membrane force (2D view), (**b**) a 3D view of the x membrane force, (**c**) top view of the y membrane force (2D view), (**d**) a 3D view of the y membrane force.

**Figure 8 materials-15-01577-f008:**
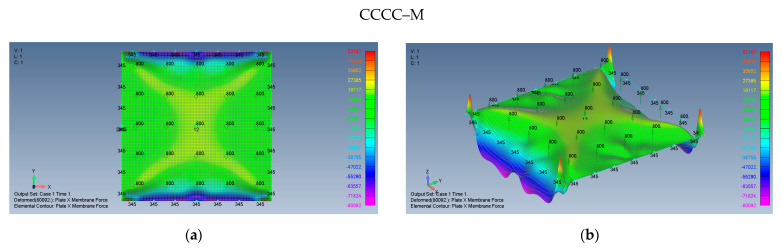
The shapes of the membrane forces for a square plate on CCCC–M boundary conditions: (**a**) top view of the *x* membrane force (2D view), (**b**) a 3D view of the *x* membrane force.

**Figure 9 materials-15-01577-f009:**
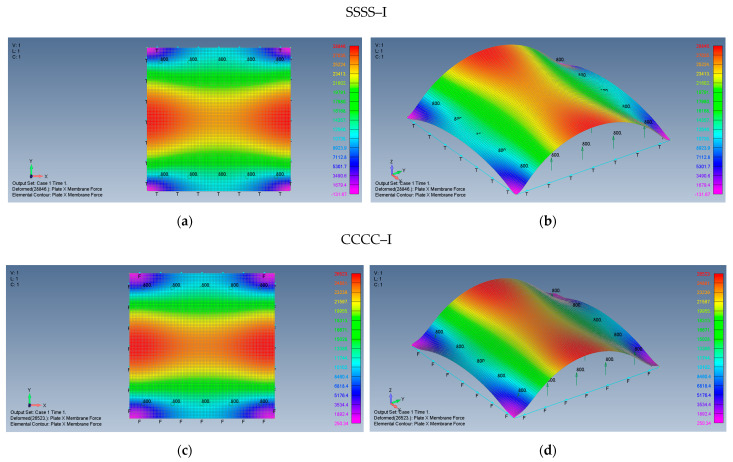
The shapes of the *x* and *y* membrane forces for a square plate on SSSS–I boundary conditions: (**a**) top view of the *x* membrane force (2D view), (**b**) a 3D view of the x membrane force and on CCCC–I boundary conditions, (**c**) top view of the *x* membrane force (2D view), (**d**) a 3D view of the *x* membrane force.

**Figure 10 materials-15-01577-f010:**
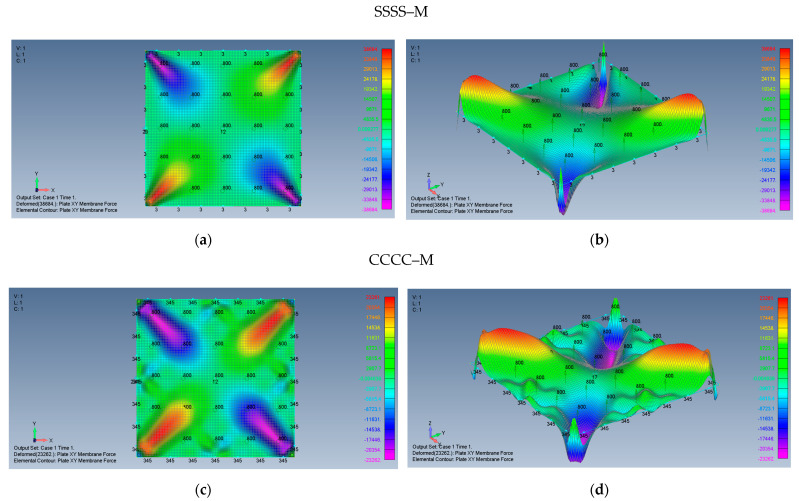
The shapes of the shear *xy* membrane forces for a square plate with movable BCs: (**a**) SSSS–M top view of the *xy* membrane force (2D view), (**b**) SSSS–M 3D view of the *xy* membrane force, (**c**) CCCC–M top view of the xy membrane force (2D view), (**d**) CCCC–M 3D view of the *xy* membrane force.

**Figure 11 materials-15-01577-f011:**
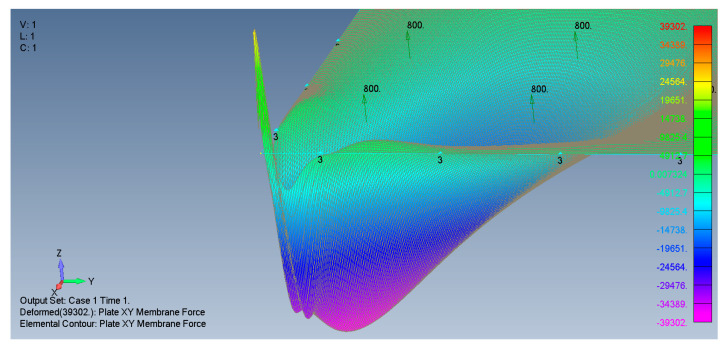
A 3D detail of the *xy* membrane forces at the plate corner for SSSS–M.

**Figure 12 materials-15-01577-f012:**
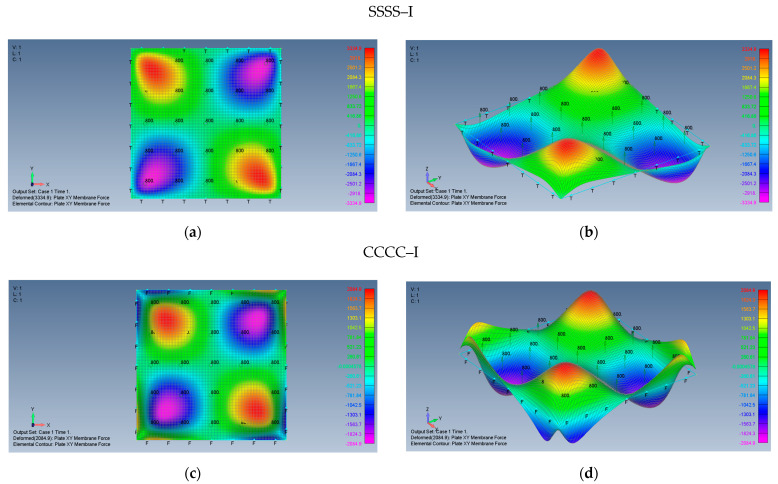
The shear *xy* membrane forces for a square plate with immovable BCs: (**a**) SSSS–I top view of the *xy* membrane force (2D view), (**b**) SSSS–I 3D view of the *xy* membrane force, (**c**) CCCC–I top view of the *xy* membrane force (2D view), (**d**) CCCC–I 3D view of the *xy* membrane force.

**Figure 13 materials-15-01577-f013:**
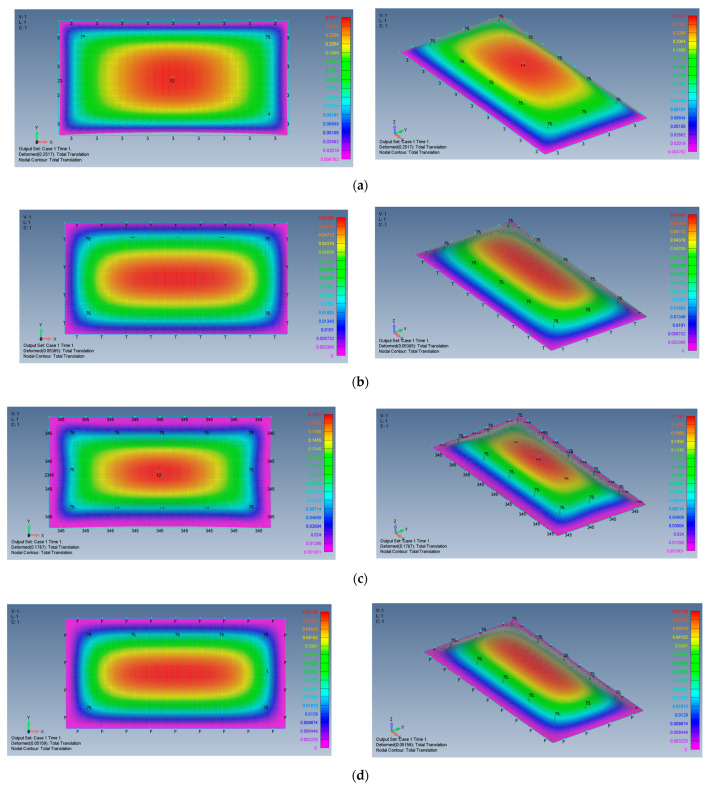
The shape of the transverse deflection for rectangular plates: (**a**) AR = 2, SSSS–M, (**b**) AR = 2, SSSS–I, (**c**) AR = 2, CCCC–M, (**d**) AR = 2, CCCC–I.

**Figure 14 materials-15-01577-f014:**
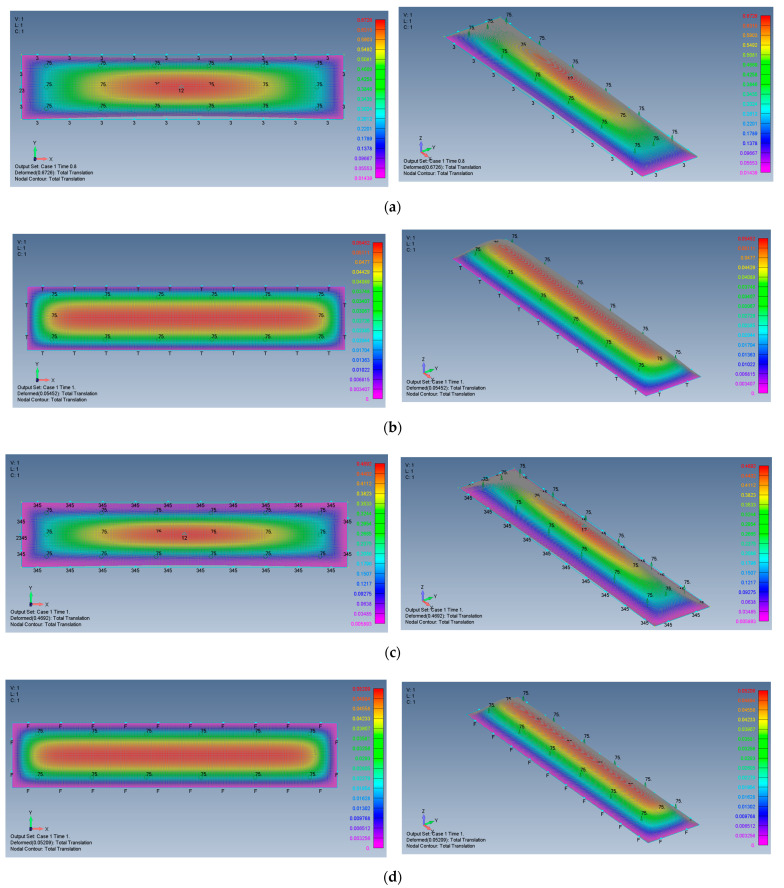
The shape of the transverse deflection for rectangular plates: (**a**) AR = 5 SSSS–M, (**b**) AR = 5 SSSS–I, (**c**) AR = 5 CCCC–M, (**d**) AR = 5 CCCC–I.

**Figure 15 materials-15-01577-f015:**
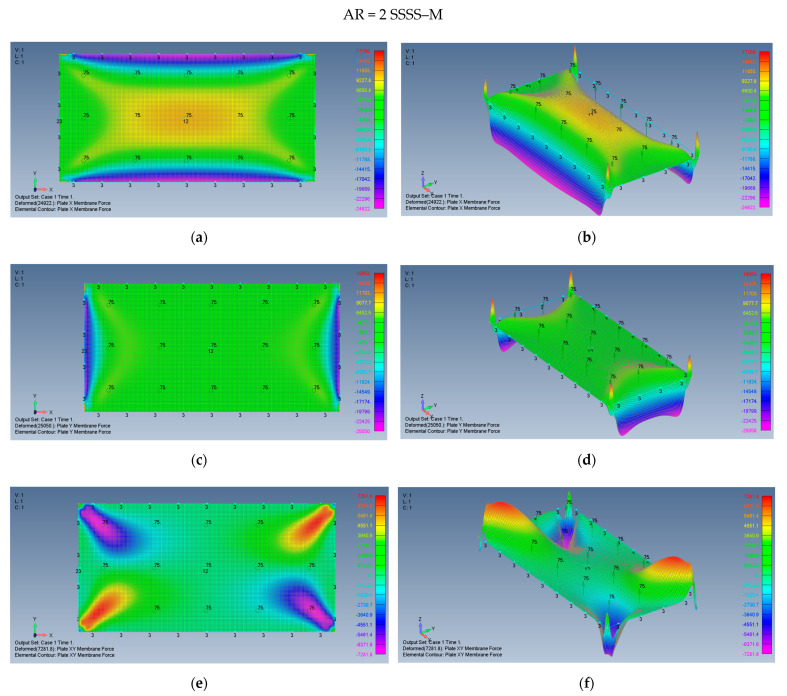
The shapes of the *x*, *y* and *xy* membrane forces for a rectangular plate AR = 2 on SSSS–M boundary conditions: (**a**) top view of the *x* membrane force (2D view), (**b**) a 3D view of the *x* membrane force, (**c**) top view of the *y* membrane force (2D view), (**d**) a 3D view of the y membrane force, (**e**) top view of the *xy* membrane force (2D view), (**f**) a 3D view of the *xy* membrane force.

**Figure 16 materials-15-01577-f016:**
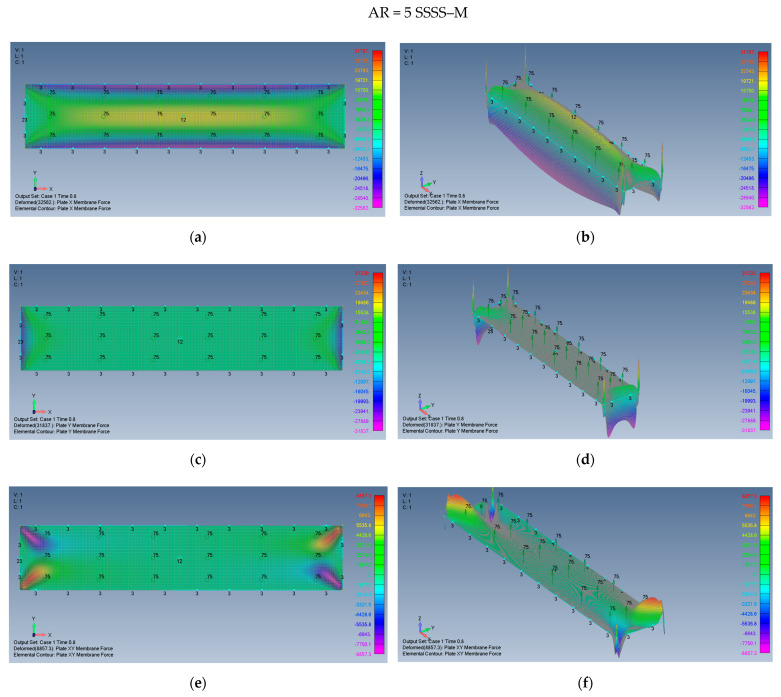
The shapes of the *x*, *y* and *xy* membrane forces for a rectangular plate AR = 5 on SSSS–M boundary conditions: (**a**) top view of the *x* membrane force (2D view), (**b**) a 3D view of the *x* membrane force, (**c**) top view of the *y* membrane force (2D view), (**d**) a 3D view of the *y* membrane force, (**e**) top view of the *xy* membrane force (2D view), (**f**) a 3D view of the *xy* membrane force.

**Figure 17 materials-15-01577-f017:**
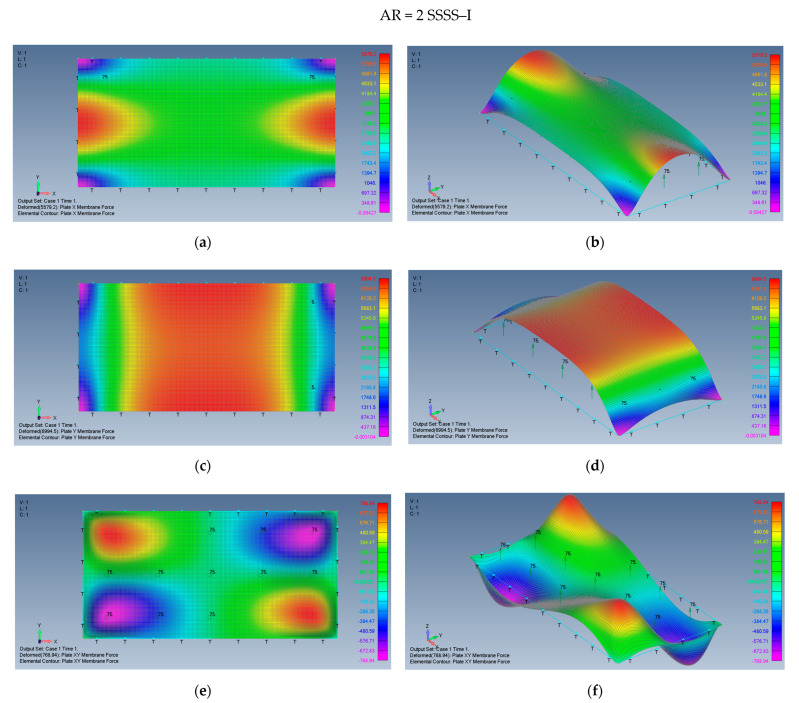
The shapes of the *x*, *y* and *xy* membrane forces for a rectangular plate AR = 2 on SSSS–I boundary conditions: (**a**) top view of the *x* membrane force (2D view), (**b**) a 3D view of the *x* membrane force, (**c**) top view of the *y* membrane force (2D view), (**d**) a 3D view of the *y* membrane force, (**e**) top view of the *xy* membrane force (2D view), (**f**) a 3D view of the *xy* membrane force.

**Figure 18 materials-15-01577-f018:**
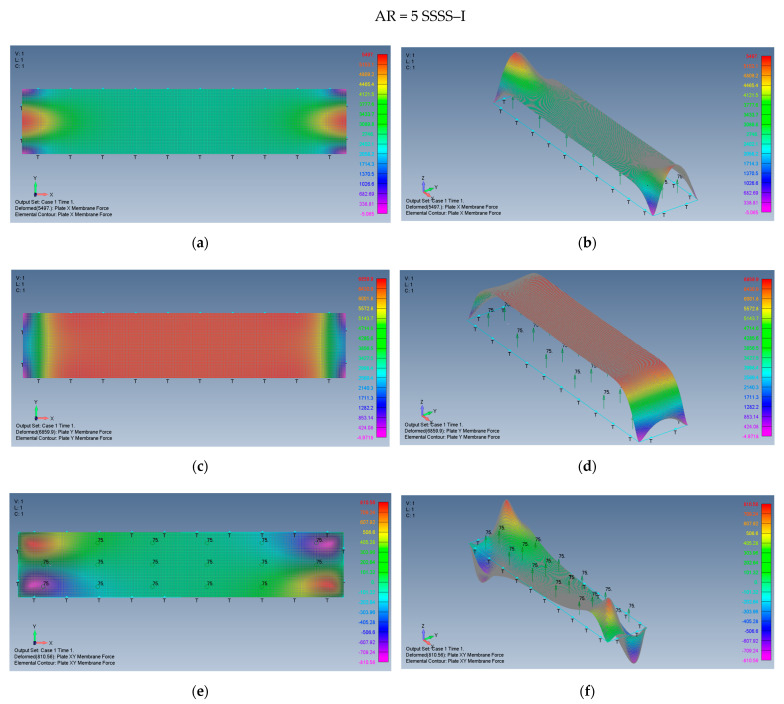
The shapes of the *x*, *y*, and *xy* membrane forces for a rectangular plate AR = 5 on SSSS–I boundary conditions: (**a**) top view of the *x* membrane force (2D view), (**b**) a 3D view of the *x* membrane force, (**c**) top view of the *y* membrane force (2D view), (**d**) a 3D view of the *y* membrane force, (**e**) top view of the *xy* membrane force (2D view), (**f**) a 3D view of the *xy* membrane force.

**Figure 19 materials-15-01577-f019:**
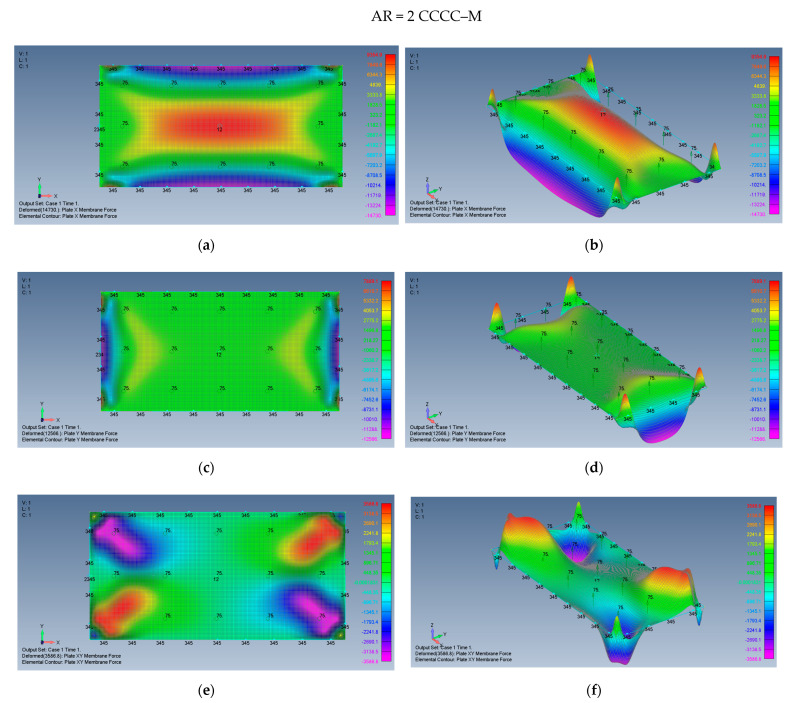
The shapes of the *x*, *y* and *xy* membrane forces for a rectangular plate AR = 2 on CCCC–M boundary conditions: (**a**) top view of the *x* membrane force (2D view), (**b**) a 3D view of the *x* membrane force, (**c**) top view of the *y* membrane force (2D view), (**d**) a 3D view of the y membrane force, (**e**) top view of the *xy* membrane force (2D view), (**f**) a 3D view of the *xy* membrane force.

**Figure 20 materials-15-01577-f020:**
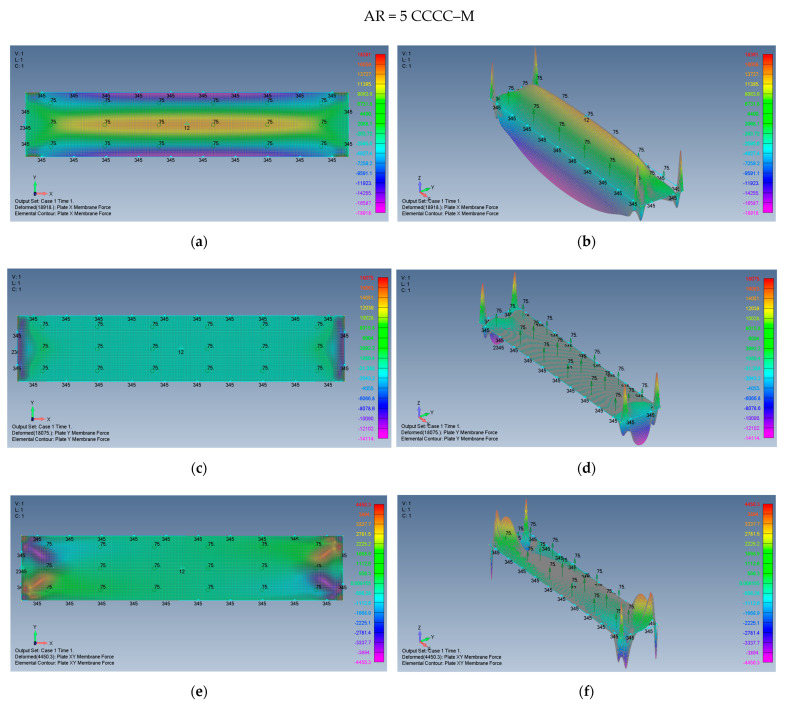
The shapes of the *x*, *y*, and *xy* membrane forces for a rectangular plate AR = 5 on CCCC–M boundary conditions: (**a**) top view of the *x* membrane force (2D view), (**b**) a 3D view of the *x* membrane force, (**c**) top view of the *y* membrane force (2D view), (**d**) a 3D view of the *y* membrane force, (**e**) top view of the *xy* membrane force (2D view), (**f**) a 3D view of the *xy* membrane force.

**Figure 21 materials-15-01577-f021:**
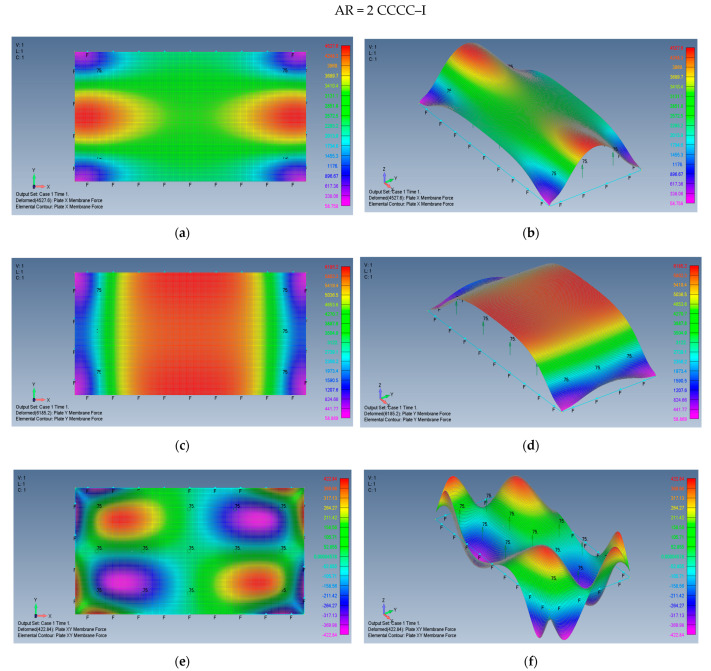
The shapes of the *x*, *y*, and *xy* membrane forces for a rectangular plate AR = 2 on CCCC–I boundary conditions: (**a**) top view of the *x* membrane force (2D view), (**b**) a 3D view of the *x* membrane force, (**c**) top view of the *y* membrane force (2D view), (**d**) a 3D view of the *y* membrane force, (**e**) top view of the *xy* membrane force (2D view), (**f**) a 3D view of the *xy* membrane force.

**Figure 22 materials-15-01577-f022:**
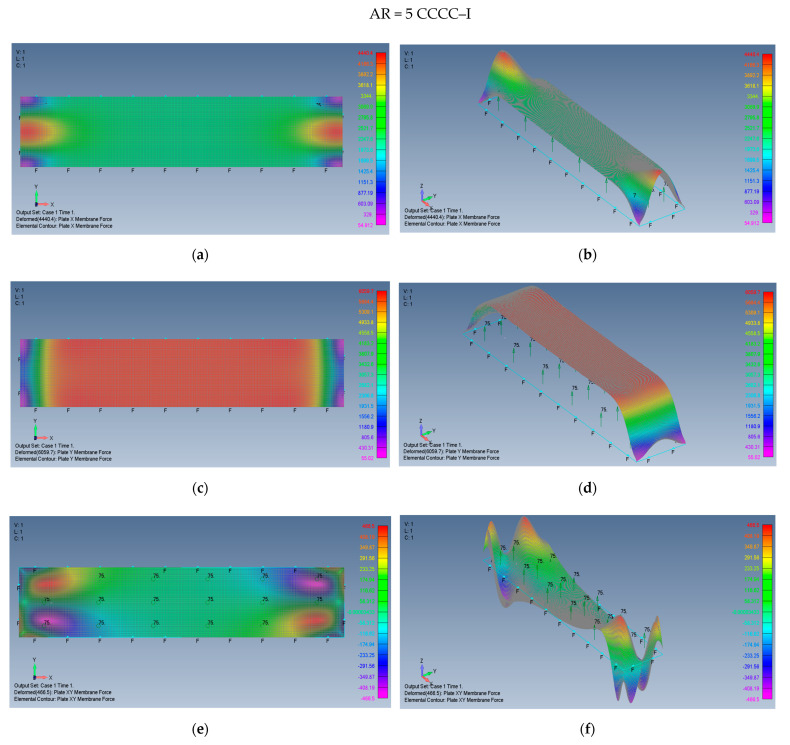
The shapes of the *x*, *y* and *xy* membrane forces for a rectangular plate AR = 5 on CCCC–I boundary conditions: (**a**) top view of the *x* membrane force (2D view), (**b**) a 3D view of the *x* membrane force, (**c**) top view of the *y* membrane force (2D view), (**d**) a 3D view of the *y* membrane force, (**e**) top view of the *xy* membrane force (2D view), (**f**) a 3D view of the *xy* membrane force.

**Table 1 materials-15-01577-t001:** Convergence Study.

Mesh Size	Number of Elements in the Mesh	Calculation Duration [min]	Mid-Point Deflection %	Max X Force %	Mid-Point X Force %
10 × 10	100	0.183	83.468%	78.516%	98.617%
20 × 20	400	0.267	93.709%	83.567%	99.647%
50 × 50	2500	0.533	99.030%	94.438%	99.988%
100 × 100	10,000	1.97	99.766%	98.249%	99.998%
200 × 200	40,000	8.92	99.942%	99.568%	99.999%
500 × 500	250,000	162	100%	100%	100%

**Table 2 materials-15-01577-t002:** Square plate—mid-point deflections.

Boundary Conditions	SSSS—M	SSSS—I	CCCC—M	CCCC—I
Mid-point deflection *w*_0_, mm	322.0	97.6	259.0	96.0
*w*_0_/*t*	26.83	8.13	21.58	8.0

**Table 3 materials-15-01577-t003:** Maximal values of the membrane stresses for various rectangular plates.

Aspect Ratio (AR)	1	2	5
Distributed Load *q* [Pa]	800	75	75
Boundary Conditions	SSSS–M		
$\sigma_{xx}$ [MPa] tensile stress @ plate midpoint	1.684	0.906	1.880
$\sigma_{yy}$ [MPa] tensile stress @ plate midpoint	1.684	0.0684	0.0126
$\sigma_{xx}$ [MPa] compression stress @ plate edges	−10.767	−2.077	−2.713
$\sigma_{yy}$ [MPa] compression stress @ plate edges	−10.767	−2.087	−2.653
$\tau_{xy}$ [MPa] shear stress @ 45° line from the corner	±3.224	±0.607	±0.738
Boundary Conditions	SSSS–I		
$\sigma_{xx}$ [MPa] tensile stress @ plate midpoint	2.095	0.295	0.217
$\sigma_{yy}$ [MPa] tensile stress @ plate midpoint	2.095	0.553	0.558
$\sigma_{xx}$ [MPa] tensile stress @ plate edges	2.404	0.465	0.458
$\sigma_{yy}$ [MPa] tensile stress @ plate edges	2.404	0.583	0.572
$\tau_{xy}$ [MPa] shear stress @ 45° line from corner	±0.278	±0.064	±0.068
Boundary Conditions	CCCC–M		
$\sigma_{xx}$ [MPa] tensile stress @ plate midpoint	1.584	0.780	1.111
$\sigma_{yy}$ [MPa] tensile stress @ plate midpoint	1.584	0.0629	0.00760
$\sigma_{xx}$ [MPa] compression/tensile stress @ plate edges	−6.674	−1.227	−1.576/+1.533
$\sigma_{yy}$ [MPa] compression/tensile stress @ plate edges	−6.674	−1.047	−1.176/+1.506
$\tau_{xy}$ [MPa] @ 45° line from corner into plate	±1.938	±0.299	±0.371
Boundary Conditions	CCCC–I		
$\sigma_{xx}$ [MPa] tensile stress @ plate midpoint	1.993	0.261	0.192
$\sigma_{yy}$ [MPa] tensile stress @ plate midpoint	1.993	0.490	0.494
$\sigma_{xx}$ [MPa] compression stress @ plate edges	2.037	0.377	0.370
$\sigma_{yy}$ [MPa] compression stress @ plate edges	2.037	0.515	0.505
$\tau_{xy}$ [MPa] shear stress @ 45° line from corner	±0.174	±0.0352	±0.0388

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
