# Peer review of "Large Deflections of Thin-Walled Plates under Transverse Loading—Investigation of the Generated In-Plane Stresses"

_materials, 2022, doi:10.3390/ma15041577_

Round 1

Reviewer 1 Report

This article investigates the large deflections of thin-walled plates under transverse loading by employing commercial software.  I cannot recommend this manuscript to publish unless the substantial revision will be done as the following:
Q1: The novelty of the present work must be significantly improved.
Q2: Rewrite the introduction., this section is mainly a list of paper summaries rather than a cohesive review of the area of research. Furthermore, the formulation should not be written in this section. 
Q3: I cannot see the detail of simulation in the software. The details of simulation must be added in the manuscript.  
Q4: The type of material which is used for structure is not clear. Please add it in the results section. 
Q5: For FE analysis, the convergence study should be done. I cannot see this section in this article. Besides the type of element should be added in the manuscript.
Q6: I cannot see the validation of present work in the results section. Please verify their numerical results with precious articles and add it in result section.  
Q7: The quality of contours in all the figures is too weak. The numbers written in the figures are not clear. 
Q8: Some results in the conclusion is obvious. Remove them please. 
Q9: How the commercial software calculates the large deflection? what was theory applied for this analysis? Please explain completely. 
Q10: Can authors explain practical application for three different boundary condition which are considered in this article?
 Q11: The literature is too weak and a few of article are reviewed by the authors. Many important research should be cited in the introduction. Some of them are as the following: 
https://doi.org/10.1080/15397734.2020.1753536  

Reviewer 2 Report

The work attempts to solve an open problem of the continuum mechanics, i.e. the stress analysis of plates with large deflections. The work regards a sensibility analysis to the plate variables in terms of boundary conditions and aspect ratio, performed by finite element analysis.

The manuscript is clear and well written. Both the methodology description and the presentation of the results need to be improved; the following points should be considered:

  • Page 5 line 194, give details about the typology of finite elements: number of nodes, degree of shape functions, number of integration points along the thickness, etc.
  • Page 5 line 196, report information about the computation capabilities of the workstation to make more consistent the comparison in terms of calculation time. Can the important difference of calculation time between the 500x500 elements model and the other ones be related to an out-of-core analysis (not enough RAM memory)?
  • The results reported in section 3 in terms of σxx, σyy, and τxy should be further presented on specific diagrams to be clearer to the reader. Figures reporting the stress components as a function of the coordinates could be useful to understand the behavior of the plate.
  • If possible, compare the numerical results obtained with finite element analysis with a reference analytical method taken from the available literature.
  • The literature about the analysis of the in-plane behavior of plates is very wide. Thus, the introduction needs to discuss more references. Among them, the following papers should be included: https://doi.org/10.1016/j.ijsolstr.2020.08.004; https://doi.org/10.1016/j.euromechsol.2020.104157

Round 2

Reviewer 2 Report

The aspects I suggested to revise have been adequately considered and discussed by the authors.

Therefore, in my opinion, the paper can be accepted for publication.